# Twenty Years of Leading the Way among Cohort Studies in Community-Driven Outreach and Engagement: Jackson State University/Jackson Heart Study

**DOI:** 10.3390/ijerph18020696

**Published:** 2021-01-15

**Authors:** Clifton Addison, Brenda W. Campbell Jenkins, Monique White, Darcel Thigpen-Odom, Marty Fortenberry, Gregory Wilson, Pamela McCoy, Lavon Young, Clevette Woodberry, Kathryn Herron, Jermal Clark, Marinelle Payton, Donna Antoine LaVigne

**Affiliations:** 1Jackson Heart Study, School of Public Health, Jackson State University, 350 West Woodrow Wilson Drive, Suite 2900B, Jackson, MS 39213, USA; brenda.w.campbell@jsums.edu (B.W.C.J.); monique.s.white@jsums.edu (M.W.); mwfort0@gmail.com (M.F.); gregory.wilson@jsums.edu (G.W.); pamela.d.mccoy@jsums.edu (P.M.); lavonyoung21@gmail.com (L.Y.); clevette.l.woodberry@jsums.edu (C.W.); katherine.herron@jsums.edu (K.H.); marinelle.payton@jsums.edu (M.P.); donna.a.antoinelavigne@jsums.edu (D.A.L.); 2Jackson Heart Study, University of Mississippi Medical Center, Jackson, MS 39213, USA; dthigpenodom@umc.edu; 3Jackson Heart Study, Community Outreach Center, Community Representative, Jackson State University, Jackson, MS 39213, USA; seainco@bellsouth.net

**Keywords:** partnership, community-based participatory research, Jackson Heart Study, engagement, recruitment and retention

## Abstract

Background: History has recorded the tremendous concerns and apprehension expressed by African Americans about participating in research studies. This review enumerates the collaborative techniques that were utilized by the Jackson State University (JSU) Jackson Heart Study (JHS) community-focused team to facilitate recruitment and retention of the JHS cohort and to implement health education and health promotion in the JHS communities. Methods: This review describes the evolution of the JSU JHS community initiatives, an innovative community-driven operation, during the period 1999–2018. Results: JSU JHS community-focused investigators published approximately 20 manuscripts, including community-led research and publications with community lead authors and co-authors, research and publications in collaboration with other JHS staff, through other JSU-funded projects. The JSU JHS community-focused unit also initiated the JHS Community Training Activities, developed the Community Health Advisory Network (CHAN), and trained and certified 137 Community Health Advisors. In addition, the JSU JHS community-focused unit developed the Collaborative Community Science Model (CCSM) that symbolized its approach to community engagement and outreach, and a Trust Scale for ascertaining African Americans’ willingness to engage in biomedical research collaborations. Conclusion: This review offers educators, public health professionals, and research investigators a useful starting point for the development, selection, or improvement of techniques to motivate, inspire, and engage community residents in a community–academia partnership that yielded maximum benefits in the areas of health education, health promotion and interventions, and biomedical research. Substantial, meaningful community engagement is possible when prioritizing elimination of health disparities and long-term improvement in health care access in the target populations.

## 1. Introduction

The Jackson Heart Study (JHS), the largest single-site epidemiological study of cardiovascular disease in African Americans, collected data from 5306 African Americans in Hinds, Madison and Rankin Counties (Mississippi). The JHS and all of its centers were funded through contracts with the National Heart, Lung, and Blood Institute (NHLBI), National Institutes of Health (NIH). Details of the JHS, its development, participants, and overall procedures are described in earlier publications [1,2,3,4]. The public health burden caused by cardiovascular disease (CVD) continues to adversely affect individuals through cost, life expectancy, medical, pharmaceutical and hospital care, particularly in the case of African Americans [5,6,7,8].

Successful community engagement is critical to realizing the JHS community-specific goals. Effective community engagement is critical to participant recruitment, participant retention, and health promotion to reduce health disparities [9,10,11,12], particularly for African Americans. Community engagement requires community partners to develop activities to enhance the quality of life for vulnerable populations, to reduce risk factors, and to decrease untimely sickness and death [13,14,15,16].

African Americans have expressed tremendous concerns and apprehension about participating in research studies. They worry about unfair treatment, exposure to harmful conditions and outcomes, and that their participation in research will not lead to improvements in health status and health care in general. Studies that are successful in recruiting African Americans are those that value and utilize the wisdom and opinions of older African Americans to heighten recruitment and retention [17,18,19]. Mistrust of medical research and scientists impedes African American participation in research studies. The United States Public Health Syphilis Study (1942–1972), also known as the Tuskegee Study, resulted in mistrust in research conducted with African Americans and discouraged African Americans from participating in research studies. As a result, some researchers hesitate to actively recruit minority participants for clinical trials because they feel that it may be difficult to obtain approval for study protocol, and African Americans who agree to participate may have higher attrition rates [20].

The primary goal of this review was to identify the collaborative techniques that were used by the Jackson State University (JSU) Jackson Heart Study (JHS) community-focused unit to develop a successful community-specific outreach and engagement approach. This review enumerates the techniques that were utilized by the JSU JHS community-focused team to facilitate recruitment and retention of the JHS cohort and to implement health education and health promotion in the JHS communities.

## 2. Materials and Methods

The evolution of the JSU JHS community-focused unit is presented as an innovative community-specific operation, that functioned during the period 1999–2018, beginning as a department in the JSU JHS Coordinating Center in the early stages of the JHS and evolving during the final contract period into a center, the Community Outreach Center (CORC) (2013–2018). The Jackson Heart Study (JHS) began with three main community-related goals to pursue successful recruitment and retention of the cohort. They are as follows:Expand minority investigator and community participation in CVD epidemiology research.Apply strategies for improving minority participation in medical research by developing partnerships between the community and research institutions.Provide health information to the African American community regarding risk factors for CVD, based on JHS findings.

Addressing the complexity of recruitment and retention in any cohort study can be a challenge for investigators and funders regarding designing relationships and partnerships to achieve sustained application. Success for the JHS required the participation of diverse stakeholders. The JSU JHS community-focused team used the Collaborative Community Science Model (CCSM) with some elements from the collaborative citizen science approach [21,22]. The CCSM is a process of customizing, arranging, and sequencing events to build community support, taking into account the unique characteristics of the local AA communities. It is a step/strategy for mobilizing community groups prior to introducing the principles of CBPR. With the CCSM, the JSU JHS community-focused team’s guidelines and action plans invited and incorporated community members’ and partners’ perspectives. The team believed that to build community support for continuing participation in the JHS exams, data collection, and its health education/health promotion agenda, engaging community members in the research process was indispensable; they could provide insight into the social and environmental conditions that affect their participation in research and their choices regarding study-related activities [23]. By including their voice, research can promote ‘knowledge democracy’ [24], a situation where knowledge is not restricted to an exclusive group of privileged academic researchers, but is freely available to all who have a desire to improve their status. This type of understanding and persistent practice strengthened the partnership between JSU community-focused investigators and the JHS community members, some of whom were JHS participants. In the relationship with the JSU JHS community-focused investigators, the JHS participants and community members were not only involved as research participants, but also as co-investigators who actively engaged in the entire research process [25,26].

Intimate collaborations were developed with more than 100 churches, community-based organizations, governmental agencies, private non-profit organizations, and other institutions of higher learning, including the University of Alabama at Birmingham, the University of Michigan, and the American Heart Association. The JSU JHS community-focused staff maintained close relationships with its community partners during its tenure as the community branch of the JHS. These relationships were important to secure a network of collaborators who could provide ongoing training for the JHS community members, the CHAN, and the JSU JHS community-focused staff and to exchange ideas that would help to build capacity in the JHS community for the purpose of eliminating health disparities and reducing the prevalence of risk factors for CVD.

### 2.1. Evolution of the JHS Community Engagement and Outreach (1999–2018)

There were three iterations of the JHS efforts to refine the principles and levels of community engagement (see Table 1 timeline).

To ensure that substantial community interaction occurred, structures and strategies for engaging the community became more advanced as the JHS community engagement evolved from the initial designation as the Community Mobilization Office (CM). The CM focused on the first original JHS community-driven goal to expand minority investigator and community participation in CVD epidemiology research. The CM was later transformed into the Community Partnership Office (CPO) that focused on the second original JHS community-driven goal of applying strategies for improving minority participation in medical research. The CPO sought to develop partnerships between the community and research institutions. The CPO was later transformed into the Community Outreach Center (CORC) that focused on the third original JHS community-driven goal of providing health information to the African American community regarding risk factors for CVD, based on JHS findings. The modifications that occurred in the structure of the JHS and the operations of the JHS centers were partly based on recommendations provided by the Observational Studies Monitoring Board (OSMB). The OSMB was installed by the funders, the National Heart, Lung, and Blood Institute (NHLBI), to provide oversight for the JHS and to provide guidance and recommendations annually on JHS operations. The OSMB functioned as an independent supervisory committee.

### 2.2. Community Mobilization (CM)

Prior to the beginning of the JHS Exam 1 in 2000, study leaders initiated the Community Mobilization (CM) office in 1999 as an office within the JHS Coordinating Center (CC), developed by JSU, with the understanding that the involvement of the local community would be central to the success of the Jackson Heart Study. Community outreach is an optimally inclusive process by which the community was provided information on the study on an ongoing basis. In addition, the JHS community-focused staff provided the JHS community with a formal structure that allowed them to provide meaningful and substantive input. Such input included community perceptions regarding participation in the study, barriers to participation, as well as community suggestions related to health promotion and education, and retention. The JHS operated with a goal to promote long-term community support for the study, while accumulating cardiovascular risk factor data about African Americans. Specific objectives identified with promoting long-term community support included 

Conducting community forums, focus groups, and community meetings to inform the community about the JHS operations;Establishing and enhancing community ties—visiting schools, churches, civic and community groups, political leaders, public health officials, etc.;Sustaining community support and involvement throughout the life of the JHS.

Recruitment barriers, such as negative perceptions of the community about underlying goals, objectives, benefits of the JHS, and distrust of the medical/scientific community, can inhibit community mobilization. It was incumbent upon JSU to identify personnel locally who possessed expertise and experience in community involvement and community relations and could develop successful strategies. These skills are critical to developing and sustaining community involvement and support.

The CM aimed to stimulate community interest and involvement in the JHS, inspiring enduring community support for the JHS, and distributing information about CVD risk factors to the community, using an extensive media campaign. The community mobilization plan was customized to special needs of the study and the community.

### 2.3. JHS Community Partnership Office (CPO)

The CM office was changed to JHS Community Partnership Office (CPO) in 2000. To improve trust, emphasis was on promoting durable, collaborative relationships, community access, relevance, and decision making. Since recruitment and retention were paramount, the CPO held information sessions for participants and their relatives, linking the participants with community health information, important community resources, and JHS progress reports. The CPO facilitated access to a reliable African American community amenable to research participation [5].

The JHS Community Partnership Subcommittee was instituted to support and provide oversight to the efforts of the CPO that was charged with building and sustaining relationships with community partners, providing heart health material to the community, conducting presentations, and participating and coordinating a broad range of community activities and events. The Community Partnership Subcommittee sought to create an environment of trust among the partners and respect for the knowledge and expertise the community brought to the table. Its work was based on the belief that the CPO agenda should do “with” and not do “for” the community. The JHS Community Partnership Subcommittee was composed of a variety of key individuals representing CPO staff members and a wide cross section of the community, whose main functions were to monitor progress of the mobilization activities, make recommendations relative to modifications in project objectives and strategies, and evaluate project directions and achievement.

### 2.4. JHS Partnership Office-Adaptation of CBPR Principles

The CPO ascribed to a community-based participatory research (CBPR) approach that promoted community involvement and ownership in all aspects of the research enterprise, including formulation of the project purpose and questions, outputs and outcomes [27,28]. The JHS viewed this arrangement as an important step to build community capacity. The principles of CBPR that were particularly important and relevant for the Jackson Heart Study were: (1) cultural relevance, where the JHS communities could define their health priorities and research agenda themselves; (2) mutual trust and respect, where the JHS and its collaborating partners established, developed, and openly cultivated respect and trust); (3) allocation of adequate and sustained resources to ensure long-term health interventions; and (4) sustainable partnerships, a long-term committed relationship between academic partners and the community partners [28,29,30,31].

### 2.5. Community Health Advisory Network (CHAN)

As health education and health promotion became heralded as important approaches for reducing risk factors for development of cardiovascular disease, the initiation of the Community Health Advisory Network (CHAN) that included JHS participants and members of the community became an important tool of the CPO. The CPO planned energetic, sustainable participant/community engagement. To ensure the continued success of the JHS and sustainability of its health education/health promotion agenda, the CPO implemented five CHANs in the JHS study area (communities of Bolton/Edwards, Mississippi, USA, Canton, Mississippi, USA, Clinton, Mississippi, USA, Pearl, Mississippi, USA, and Jackson, Mississippi, USA) and organized health promotion/health activities to train the CHAN members to acquire the capacity to facilitate the JHS goals of reducing health disparities and promoting health education within their respective communities [32].

All of these CHANs worked to improve heart health conditions in their community. They organized health fairs, blood drives, participated in community walks, presented forums on heart disease-prevention topics, conducted heart health sessions for children and youth. Most of the CHAs received 30 h of training in the areas of cardiovascular health and leadership development, and participated in training in a variety of topics along with other community partners to build capacity to improve the quality of life in their respective communities. In partnership with the University of Michigan, Ann Arbor, the JSU JHS community-focused team provided training to its partners that included community organizations, academia, policy makers and students on community-based participatory research, social determinants of health, and policy advocacy (Table 2).

### 2.6. Training Credentials of Staff and CHAN Members

CPO staff and community members participated in training conducted by the master trainer for the CHAN curriculum, in collaboration with the University of Southern Mississippi (USM). In addition, all CPO staff and Community Health Advisors (CHAs) received training from practitioners in the relevant topic areas; the Genetics component was taught by a geneticist, the Heart session was taught by a cardiologist and nutrition information was taught by a dietician. The leadership development components were taught by the CPO staff. Everyone received certification from the Center for Sustainable Health Outreach at USM. Other training included certifications in community outreach training, leadership development training, and physical education training.

The CHANs developed and implemented new health and wellness projects throughout the cohort community. The new projects met and exceeded the original goals of the CHAN program. These goals were:Promote awareness of Jackson Heart Study initiatives;Inform the lay community of Jackson Heart Study findings;Provide culturally appropriate heart health information to the community;Provide venues for health screenings, exercise demonstrations, and health care information exchanges;Increase the number of community agencies, churches and health care providers who are Jackson Heart Study partners;Introduce healthy lifestyle choices to the public; andProvide leadership training to Community Health Advisors (CHAs) to facilitate financial independence and sustainability.

The CHAs collaborated with various community-based health care providers in each community, such as community health centers, hospitals, schools/school systems, City Park and Recreational Departments, ambulance services, churches, libraries, nursing schools, police departments, community action agencies, city government, YMCAs, community health and wellness groups, and colleges/universities.

### 2.7. Community Outreach Center (CORC)

The Community Outreach Center (CORC) represented the final stage of the community engagement and outreach evolution (Table 1). During the final stages of JSUs involvement in the JHS community outreach and engagement endeavor (2013–2018), UMMC was awarded the JHS Coordinating Center (CC), originally developed and operated by JSU, and a brand new Community Outreach Center (CORC) was created and allocated to JSU. For CORC, building and maintaining a trusting environment and a sense of caring were essential elements to ensuring community support and continued involvement in JHS activities. CORC, housed within JSU’s School of Public Health, was one of five new JHS Centers. It evolved from the CM and the CPO, former divisions of the JHS Coordinating Center that was developed and operated by JSU from 2000 to 2013. CORC, intent on adhering to the CBPR principles, continued to build long-term, health promotion partnerships with diverse groups of community representatives. To facilitate its activities with the community, CORC investigators developed the Collaborative Community Science Model (CCSM), also known as the AJA Six-Facet Model of Community Engagement (Figure 1) [33]. CORC contributed to the JHS ability to achieve its aims by using the medium of community health education activities and disseminating health promotion and prevention messages in the Jackson community. The five specific aims for CORC were:Plan and conduct community outreach activities and events to communicate study findings and health promotion messages, promote the JHS, retain study participants, and build trust in the community;Administer the JHS Community Health Advisors Network (CHAN) to promote cardiovascular disease prevention in the Jackson area through health education programs and other activities as appropriate;Convene a JHS Community Ethics Advisory Board to advise the JHS on ethical issues concerning participation in research, which me twice per year or more frequently as needed;Publish a newsletter every six months for JHS participants and the community reporting research findings, health promotion messages, and other appropriate items of interest;Participate in JHS initiatives and committees, manuscript development, and provide support for data analysis.

### 2.8. CORC’s Community Training Activities

CORC’s community training programs involved seminars conducted by trained professionals, some of whom were JHS Staff, to build capacity among community members who had an interest in learning how to eliminate or reduce health disparity and CVD (Table 3). The participants were exposed to Community Health Awareness Training, General Health Education Training, CHAN Training, Know Your Numbers Training (KYN), and CHAN Sustainability Training, while receiving informal counseling and social support. They attended activities monthly to build capacity and promote healthy lifestyles, behaviors, and personal responsibility [32].

### 2.9. CORC Community Research Agenda

CORC staff developed a community research agenda with the community partners promising to:Include CHAs on appropriate CORC working groups for manuscript development to address their particular community’s health concerns and explorations of solutions to selected health disparities;Facilitate CHA development and conduct of scheduled presentations for community members;Continue a more robust effort to facilitate CHA documentation of all activities conducted;Continue to strongly encourage CHAs to keep records of guest speakers who provide presentations/updates that are associated with JHS research findings;Develop ideas and support policies that address behavioral changes that are based on scientific evidence provided from JHS findings and/or related study findings; andEncourage CHAs to participate in a newly created community-based participatory working group within the CORC.

## 3. Results

The JSU JHS community-focused efforts by the CM team resulted in many salient accomplishments that not only promoted recruitment and retention of the JHS cohort, but also embraced a solid capacity-building agenda in concert with the JHS community. The JSU JHS CORC facilitated the development of the JHS Community Advisory Coalition/Board, a partnership comprised of citizens of the tri-county area of Hinds, Madison, and Rankin counties to represent and promote the perspectives of the community in the conduct of the JHS; community perspectives were obtained from state and community organizations, health and medical educators and providers, and JHS participants. The JSU JHS CPO that preceded the JSU JHS CORC created supportive spaces within the CHANS to enable the promulgation of sustainable community groups. The breadth and depth of information sharing through a community network play a significant role in modifying negative attitudes, behaviors, and practices among community partners, which serves as an important requirement for the community network to expand and to promote its healthy development.

Following training under CBPR renowned pioneers, Barbara Israel and Amy Schulz and others from the UM School of Public Health Center for Integrated Approaches for Health Disparities (CIAHD), the JHS Community Health Advisors contributed towards the development of two fact sheets. They developed “Social Determinants of Health” and “Food Access in Mississippi” as an end product of their training.

Through CORC’s collaboration with the University of Alabama, Birmingham, CORC community partner, Dorothy McGill, developed and submitted a pilot research project proposal, “Uniting Neighbors to Inspire Today’s Youth” (UNITY), which was funded. The purpose of the study was to test a social media video message among young African American girls of ages 10–13 on healthy eating/good food choices. The activities included focus groups, trips to the farmers market to view, purchase and learn about fruits and vegetables that are not a mainstay of their eating habits, organized physical activity and the creation of a video highlighting the activities. A poster on this project was presented by community partner, Dorothy McGill. Three video messages were produced on nutrition, physical activity and chronic diseases. These messages can be incorporated in health education classes, shared on social media outlets popular with youth such as Snapchat, Instagram, and YouTube to educate other middle school students on the importance adopting healthy nutrition. A poster on this project was presented by community partner, Dorothy McGill, at the American Public Health Association 2017 Annual Meeting.

CORC’s engagement of CHAs in the research process resulted in the research study, “Step N2 Life: A Pilot Investigation on the Benefits of Community-Developed Intervention to Reduce Hypertension in Mississippi”. It was conceived by community partners and conducted in two Jackson neighborhoods to reduce hypertension and its risk factors in African American females. The study was funded by a small ($1000) grant from the Mississippi State Department of Health to CHAs/community research fellows (CRFs). Findings from the study were published [33].

Findings from the Community Health Awareness Knowledge Change Survey administered in collaboration with the Jackson State University Center of Excellence in Minority Health and Health Disparities (CEMHD) revealed that the majority of the participants (86%) believed that the CORC’s health education and promotion activities helped to improve their personal health and quality of life. Approximately 84% of them acknowledged that their involvement in these activities served to increase their knowledge of cardiovascular disease and its risk factors in order to improve the quality of life in their community.

CORC’s continuous interaction with the community partners played a major role in the retention of the JHS cohort, the highest retention rate of any national cohort study. By 2018, approximately 123 or 90% of the 137 CHAs were actively engaged in their respective CHANs, the five-group network that is indigenous to Hinds, Madison and Rankin counties.

### 3.1. JSU JHS Community REACH—Using Enhanced Communication Pathways

#### 3.1.1. JHS Heartbeat Newsletter

The JHS *Heartbeat* newsletter was published twice annually to disseminate study findings and provide CVD risk factor reduction and prevention information to all JHS participants and the metro Jackson community. It also highlighted the research and activities of scholars, JHS investigators and other researchers. The JHS Heartbeat was a JSU/community-managed publication that the JHS used to reinforce health awareness and understanding of cardiovascular disease among the community. The JHS Heartbeat was also used to inform the cohort and community about JHS programs, events, and initiatives. Additionally, the newsletter facilitated communication among JHS staff, investigators, cohort members, contractors and the extended JHS family. The JHS Heartbeat was mailed to the cohort two times a year.

#### 3.1.2. Annual Community Events

The JSU JHS community-focused team planned and implemented an agenda that included several annual events. All events were held at the Jackson Medical Mall, Jackson, Mississippi, the location of the Jackson Heart Study. The Jackson Medical Mall is also recognized as a public walkway/exercise site for local residents and offers a community room for community members. The major activities included 

(1)The Annual Community Monitoring Board Meeting was held in June each year. The Community Monitoring Board, also known as the “State of the Study”, was specifically designed to obtain suggestions, recommendations and pose questions from the community, as well as share information on the “State” of the Jackson Heart Study. It was a vehicle created by CORC to maintain the community’s involvement as a research partner, and to sustain community and cohort support throughout the duration of the study.(2)The Annual Birthday Celebration, also called the JHS anniversary, was an event that commemorated the initiation of the JHS. This event was held on the last Saturday in September every year and featured an agenda that included featured speakers on CVD-related topics, as well as health screenings. Each birthday celebration had a specialized theme selected by the JHS community, and the featured speaker’s message focused on the theme.(3)The Annual Celebration of Life was held each year in February as a mark of celebration of “Black History Month”, and as an opportunity for the JHS community to express their opinions and concerns about the JHS and provide ideas for the future direction for the study.(4)The CHAN Annual Health Awareness Day event was organized by the CHANs.(5)Community Health Advisors (CHA) held annual “Retention Holiday Gatherings” in their community, providing gifts for needy residents and fellowship during the Christmas holiday season.(6)The annual book-bag giveaway is another event organized by the CHAN where book bags with school supplies were donated to families with children in need.

While the CHANs actively participated in organizational meetings and planning of other events of the JSU JHS community team, JSU JHS staff provided technical assistance to CHANs to facilitate their events.

#### 3.1.3. CORC-Sponsored Community Training Events

In partnership with the University of Michigan (UM), Ann Arbor, CORC provided training to community organizations, academia, policy makers and students on community-based participatory research, social determinants of health, and policy advocacy. Collaboration with UM led to community partners becoming more engaged and vocal on policy development in their communities. The impact of the trainings conducted by the UM, beginning in 2013 and ending in 2017, encouraged residents to advocate on matters affecting their local communities. They learned to identify issues, develop and implement strategies to address those issues, and strengthen relationships with local policy makers to effect change.

Collaboration with the American Heart Association (AHA) led to the development of the AHA HBCU Undergraduate Scholars Program. CORC’s Principal Investigator was appointed to the AHA Greater Southeast Board, the Metro Jackson Board, and served as chair of the Multi-Cultural Initiatives Committee and the local Affiliate. JSU was named the lead HBCU Institution. Several JHS investigators were engaged to serve as mentors to students participating in the program. The impact of the creation of the AHA HBCU Scholars Program was expected to lead to a growing interest in public health and cardiovascular research among undergraduate students. The program offers training opportunities to scholars from Fisk and Tennessee State University, Tennessee, to Jackson State University and Tougaloo College, Mississippi, and Myles College, Alabama.

The following is a partial list of other community organizations, academia, policy makers and students who participated in CORC-sponsored trainings: (1) Health Ministries/Local Churches; (2) JSU School of Public Health Graduate Students; (3) JSU School of Public Health Faculty; (4) Hinds Behavioral Health; (5) Innovative Behavioral Services; (6) UMMC Faculty; (7) City of Bolton; (8) Spectrum Employment; (9) Metro Jackson Prevention Coalition; (10) Mississippi Department of Health; (11) Mississippi Urban Research Center (MURC); (12) Hinds County Government; (13) My Brother’s Keeper; (14) Bolton/Edwards CHAN; (15) Jackson CHAN; (16) Clinton CHAN; (17) Canton CHAN; (18) Rankin County CHAN.

#### 3.1.4. Other Notable CORC Accomplishments and Networking Avenues

The CORC PI was appointed by the Governor of Mississippi to serve on the Mississippi Tobacco Control Advisory Council. The Council advises the Mississippi Department of Health Office of Tobacco Control on the development and implementation of tobacco-related programs and activities. New relationships have developed as a result of this appointment at the state level and with non-profit organizations working to reduce/eliminate the use of tobacco, e-cigarettes, and other substances impacting cardiovascular and other health outcomes.

CORC’s PI was among a select group invited to participate in the National Institutes of Health’s (NIH) two-day Precision Medicine Initiative (PMI) Participant Engagement and Health Equity Workshop and was appointed a member of the PMI protocol development team. Precision Medicine Initiative (PMI) is a new research effort to transform strategies used to improve health and treat disease that will work best for individual patients. CORC’s PI and Senior Research Scientist represented the JHS in a working group on multi-level interventions research that targeted hard-to-reach, high-risk or vulnerable populations and communities. This working group was convened by the National Heart, Lung, and Blood Institute (NHLBI) and included the top experts in the field. CORC initiated the JHS Ethics Board that provided assurances that the JHS participants’ and community’s rights would be respected, honored, and protected. In addition, CORC investigators developed the African American Trust Scale, a questionnaire to accurately measure the feelings of trust and security in research-related environments.

#### 3.1.5. Media Connection

CORC retention strategies included multiple media pathways (radio, newspaper, television). CORC staff also participated in educational sessions and dissemination of JHS findings. CORC engaged Jackson State University Communications; Jackson State University radio station (WJSU FM 88.5), WKXI/WOAD, WMPR, WRTM and newspapers, the Clarion Ledger, the Jackson Advocate, and the Mississippi Link newspaper. The Jackson Advocate made an early commitment to support all CORC’s activities and provided extensive coverage of the annual events to its readership.

#### 3.1.6. Significant CORC Scientific Contributions

The Collaborative Community Science Model (CCSM), also known as the AJA Six-Facet Model of Community Engagement, developed by CORC investigators, promotes community participation in research that explores human conditions and public welfare to increase community knowledge to initiate change. The CCSM customizes and arranges the sequence of events to positively build community support, taking into account the unique characteristics of the local AA communities. The CCSM advocates that, in a community–academia partnership, average community members with varying backgrounds should have the opportunity to work side by side with academic researchers and practitioners to identify and answer relevant research questions relating to the welfare of communities (Figure 1).

## 4. Discussion

### Lessons Learned, Challenges, and Recommendations for Improvements

Important lessons learned during the JSU JHS community-focused periods fall into two categories: (1) fulfilling JHS obligations, and (2) maintaining successful community engagement and assuring sustainability. The CM and CPO periods were characterized by successful recruitment and retention efforts. The CPO period was synonymous with the development of the CHANS and the acceleration of health promotion activity. The CORC era was characteristic of the efforts to build capacity among the JHS communities with increased collaboration with university partners for training and professional development and engagement of community partners in research activities. With respect to JHS CORC obligations, 2013–2018 was a very challenging period, handicapped by limited distribution of resources—personnel, institutional, and organizational support. Had it not been for a dynamic staff that was totally committed to making sacrifices for the success of the study and the benefit of the community, regardless of their own personal losses, maintaining a capable, proficient staff and successfully completing and exceeding all proposed objectives and expectations with the existing limitations would have been unattainable. The JHS would have achieved more success in maintaining scientific integrity if the decisions that were made had the benefit of input from the experts who were actively and directly involved with the center’s operations on a daily basis. Feedback from the community recommended that the JHS do more to develop ideas and support policies that address behavioral changes that are based on scientific evidence provided from JHS findings and/or related study findings. In addition, more respect for the voices and input of the community members could contribute to sustainability.

Another important lesson learned from this experience was that institutions agreeing to enter into the types of subcontracts mentioned above must strive to ensure that institutional policies and other barriers do not hinder and/or discourage non-employees (community- and faith-based individuals and partners from other universities and organizations) from participating in these opportunities, especially since the goal is to increase the knowledge of and improve health outcomes of community partners and other underserved populations. It is noteworthy that, as a result of the training that they received, many of the CORC community partners developed the capacity to share knowledge and disseminate health information in communities through presentations, manuscript development, and proposal writing, and they feel that they have played a major role in reducing negative risk practices that have the potential to add to the prevalence of cardiovascular disease.

In the nearly two decades that Jackson State University has led the Jackson Heart Study community activities, the funding agency did not provide adequate/appropriate funding commensurate with the significant role and importance of community to this study. The funders should consider supporting community members who participate voluntarily by providing adequate and appropriate funding in the budget. The funding allocated for managing these critical activities did not demonstrate adequate appreciation or acknowledgement of the value of community for their hard work and commitment, nor did it adequately facilitate and promote sustainability. Resources provided for the operations expected have not been adequate for the sacrifices and contributions the community members have made to the success of this study over the past 20 years. There were very minimal funds available to support the efforts of CHAs who served in lead roles for their local CHAN. Each received a small stipend over the five-year contract period. Over this five-year period, community volunteers and CORC staff used personal funds to support many of the JHS CHAN events/activities to ensure sustainability. It was only because of the trust the community had in the CORC staff at Jackson State University that CHAs continued their involvement with the study. This is evidenced from anecdotal, qualitative and quantitative data collected over the years. Continued lack of support for community members and activities could ultimately lead to the demise of community participation and support and other volunteer support of the JHS.

Even though it is fashionable for research organizations and senior executives to boast on paper and in public about being transparent, applying CBPR principles, and inviting and accepting input from community members as equal partners in the organizational and management oversight of research operations, the actual research landscape would reveal a much more divergent image of the decision-making environment. The individuals directly responsible for promoting and managing the ideals of CBPR would caution that the path to achieving research communities’ autonomy and opportunities for representation is not an easy path, and often involves discord and friction. The struggle for autonomy and opportunity for community representation often involves resistance and conflict before parity can be visible and the voices and opinions of communities can be respected and accepted. One of the reasons for the successes of the JHS community engagement and outreach operations managed by CORC is the obstinate persistence of the JSU research contingent, as they assumed the role of advocates for those community members who ordinarily would not have a voice at the decision making.

Barriers encountered have been overcome by targeted JSU JHS staff communication and debriefings, implementation of specific community activities to engage and include community members, awareness and sensitivity to the cultural and ethnocentric needs of the community, and the willingness to take time to explain study activities to participants and community members in a warm, respectful, and friendly atmosphere. The JSU JHS efforts on retention is a model for retention of African Americans in research. Similar retention activities can be applied by other researchers in minority settings and those investigators conducting research involving other racial/ethnic groups.

In the spirit of CBPR, JSU JHS investigators combined personal qualities with relevant knowledge and experience to help participants and community members form opinions and make decisions regarding their continued participation in JHS health promotion and research activities. Through the established trusting, bonding relationship, the community partners understood that they were always afforded opportunities to interpret information to help them make the right choices for themselves and their families. To the JHS community members, their voice that echoed their emotions and attitudes became an important component of the decision-making process and an important tool of the JSU JHS team.

## 5. Conclusions

The main objective of this paper is to outline the activities that were conducted by the JSU community-focused unit and the direct impact these activities had on cohort accrual rates and subsequent retention. The name change throughout the years from 2000 to 2013 reflected modifications to the mission and objectives based on ongoing assessments/evaluations and recommendations from the OSMB. These modifications described in this account had the ultimate goal of perfecting the craft of recruitment and retention. The major goal of all of the activities of the JSU staff was to stimulate recruitment and retention in order to facilitate a successful research enterprise.

This account sought to provide the historical context for the successes of the JHS recruitment and retention strategies from 2000 to 2018 that was spearheaded by Jackson State University, and this historical account includes the JHS community outreach and engagement accomplishments that were highlighted in previous accounts. The impetus for the unit was to add to the science of community outreach and engagement and to provide current and prospective community engagement and outreach enthusiasts with a comprehensive account of the strategies used to build trust in an African American community with regards to participation in research and to align the interest of the recruited cohort with the commitment to ensure a successful research experience.

The work of the CORC is credited with facilitating recruitment of the largest cohort of African Americans to participate in a study of CVD. The CORC efforts to maintain a viable cohort were successful in securing the highest retention rate among NIH-funded studies. Eighty-five percent of the cohort returned from Exam 1 to Exam 2. CORC, as the first and only center dedicated to community outreach and engagement in a national cohort study funded by the NHLBI, NIH, stands alone as the standard bearer and the model for inspiring and motivating African Americans to become committed and engaged in academic-community research activities [34]. The primary goal of this review was to identify the tools that were used by CORC to develop a successful community-driven engagement approach. CORC’s programs were designed to inspire disease risk factor prevention and chronic disease reduction among the JHS community by inspiring the community to propose policy recommendations. This review offers educators, public health professionals, and research investigators a useful starting point for the development, selection, or improvement of techniques to engage community residents in a community–academia partnership. Substantial, meaningful community engagement is possible when prioritizing elimination of health disparities and long-term improvement in health care access [13].

The accomplishments of CORC lead to important implications for reducing urban and rural health inequities. Even though research has reported some degree of progress in the battle against CVD, it is still a major health concern, accounting for approximately 1 million deaths in the United States each year [35]. CORC investigators have identified some optimal strategies to build community determination and support to universally address social inequities in health in Mississippi. To reduce racial inequities in health, particularly in African American communities, investigators and researchers should use CORC’s experiences and take action based on the successful efforts to recruit and retain the largest cohort of African Americans in any research study. This includes sustaining initiatives to raise awareness in study communities about the prevalence and risk factors of diseases and the need to maintain participation. They must enhance the capacity of individuals and communities to actively participate in intervention efforts. Communities must implement large-scale efforts to eliminate the ideologies and stereotypes that initiate and sustain inequities and health disparities.

Interventions to manage CVD risk factors, such as diabetes, obesity, hypertension and sedentary life style, have been successful. Community-based participatory research (CBPR) approaches in faith-based settings have had measurable benefits over varying periods of time. Addison et al. (2016) documented selected CORC strategies for building collaborative health promotion partnerships. CORC incorporated principles of CBPR to create an environment of trust; provide opportunities for substantive community involvement in all phases of development and implementation of the JHS; conduct health promotion/education activities; and educate the community on the progress and processes of the JHS. Innovative CORC strategies included a two-day strategic planning retreat that comprised community groups; the CCSM, also known as Addison-Jenkins-Antoine-LaVigne (AJA) Six-Facet Model of Community Engagement; and a Logic Model for Effective Academic-Community Partnerships. Addison et al. (2016) concluded that the AJA model and the logic model are useful to stimulate community-engaged research and community outreach for health education and health promotion [36].

All of the activities of CORC facilitated through JSU have strongly supported Objective #1 of NHLBI Strategic Goals and Objectives which emphasizes that “Understanding normal biology is the backbone of all biomedical science. It is essential for understanding homeostatic maintenance, predicting how biological systems respond to their environment, and recognizing disease and targets for intervention”. The expected outcome of Objective #1 is to “generate a better understanding of how environmental exposures, social determinants, and behaviors (e.g., diet and physical activity) modulate biological systems to sustain health and promote resilience”. From 1999 to 2018, community engagement promoted by CORC included community mobilization; developing and maintaining community partnerships; creation of the Community Health Advisor Networks; conducting presentations at community- and faith-based gatherings; creation and management of the Community Ethics Advisory Board; and communication with the community via a JHS participant and community newsletter, as well as multiple public media pathways.

Planning, implementing and evaluating CORC’s actions to promote cardiovascular health have evolved from a focus on mobilization and partnership, to community outreach and community engagement. Engaging the Jackson area community in all phases of the JHS was a mutually valued practice by JHS investigators and the JHS community. CORC’s rich background of knowledge, abilities, interests, experiences, lessons learned, and a longstanding trusting relationship with community were reflected in specific approaches for addressing CORC’s objectives for the period 1999–2018. These objectives facilitated realization of Objective #3 of NHLBI Strategic Goals and Objectives which is to “investigate factors that account for differences in health among populations because a wide range of behavioral factors and socioeconomic inequities also contribute to health disparities”. CORC’s role to help the JHS “better understand the causes of population health differences” also identified “strategies to effectively address these differences”. JHS CORC facilitated investigations ranging from public health to community-centered implementation research to reduce health disparities. With community engagement being a key tenet of translational research, and equity being a core value of community-based participatory research, demand has grown for skills, knowledge, training, and strategies to enhance multi-disciplinary partnerships in the research and practice of public health [37]. The partnerships developed by CORC have benefitted science by generating a research-to-practice platform that is informed by theory and the needs of the JHS communities. These partnerships have generated interventions that are context sensitive and culturally congruent [38].

## Figures and Tables

**Figure 1 ijerph-18-00696-f001:**
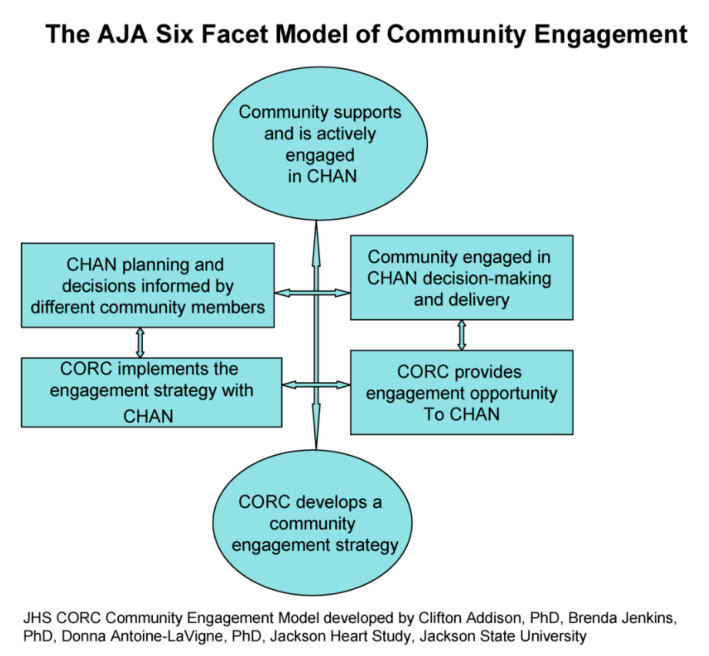
Community Engagement Model.

**Table 1 ijerph-18-00696-t001:** Evolution of the JSU JHS Community Engagement and Outreach (1999–2018).

Official JHS Designation	Primary Responsibilities	Tools/Strategies Implemented
Community Mobilization (CM) (1999–2000)	Planning, organizing, and managing focus group, media outreach and community outreach and communication.	Informed the Jackson metropolitan community about the initiation of the JHS. Created community awareness of the JHS, and notified local communities about the Study’s benefits, eligibility for participation, upcoming events, and agendas
Office of Community Partnership for Awareness Education (CPO) (2000–2013)	Harmonizing the community mobilization-implementation processes, publicizing efforts, seminars, conferences, focus groups, and public relations.	Built alliances between the community and the Study; built and sustained community trust in the JHS; identified and supported best practices to address challenges and barriers, all grounded in the principles of CBPR; channeled community messages through local churches to ensure community awareness about the JHS, its benefits, and JHS cohort eligibility and the recruitment process; sponsored 3 annual community events-Celebration of Life, Community Monitoring Board Meeting; Birthday Celebration.
Community Outreach and Engagement Center (CORC) (2013–2018)	Promoting a friendly, courteous, and shared community–academic setting that stimulates health equity through prevention, education, training, and research.	Organized CHANs to improve the status of others with health-related issues, social services and quality of life; promoted long-term sustainability and perpetuation of healthy lifestyle and heart healthy practices through dedicated community drive to provide complimentary health education; continued operating 3 annual community events -Celebration of Life, Community Monitoring Board Meeting; Birthday Celebration.

**Table 2 ijerph-18-00696-t002:** Areas of Training for Members of the JHS CHANs and Community Partners.

Community Organizations, Academia, Policy-Makers Participating in CPO-Sponsored Trainings	Topics Covered in Training Cardiovascular Health
Health Ministries/Local Churches	Genetics
JSU School of Public Health Graduate Students	Heart Disease Prevention
JSU School of Public Health Faculty	High Blood Pressure
Hinds Behavioral Health	Obesity
Innovative Behavioral Services	Cholesterol
UMMC Faculty	Physical Inactivity
City of Bolton	Heart Healthy Eating
Spectrum Employment	Stress Reduction
Metro Jackson Prevention Coalition	Smoking Cessation
MS Department of Health	
Mississippi Urban Research Center (MURC)	**Leadership Development**
Hinds County Government	Community Assessment
My Brother’s Keeper	Group Problem Solving
Bolton/Edwards CHAN	Health Promotion
Jackson CHAN	Helping Interventions
Clinton CHAN	Consensus Decision Making
Pearl/Rankin County CHAN	Effective Communication
Canton CHAN	Community Action Planning
	Workshop Method
	Coalition Building

**Table 3 ijerph-18-00696-t003:** Jackson State University Jackson Heart Study (JHS) Community Outreach Center (CORC) Trainings 2013–2017.

Name of Course	Course Description/Purpose
Collaborative Research Capacity-Building Training for Community-Based Organizations	Course focused on understanding of collaborative research and how it can benefit community-based organizations, knowledge for deciding whether and how to engage in a research partnership, and becoming familiar with community-based participatory research—an approach to collaborative research.
Introduction to CBPR: Rationale, Principles, Partnership Development and Examples	Course focused on CBPR definition, key principles, rationale for developing a CBPR approach for health, describing a process for creating, maintaining a CBPR partnership: how to bring together partners, identify assets, select priority issues.
Conducting Etiological and Intervention Research using CBPR	Course focused on benefits of conducting CBPR for community- based organizations and Healthy Environments Partnership: Cardiovascular Health in Detroit, Creating a Smoke-free Jackson, MS case studies.
Introduction to Collaborative Research for Community-Based Organizations	Course focused on understanding of collaborative research and how it can benefit community-based organizations, knowledge for deciding whether and how to engage in a research partnership, and becoming familiar with community-based participatory research—an approach to collaborative research.
Introduction to CBPR: Rationale, Principles, Partnership Development and Examples	Course focused on CBPR definition, key principles, rationale for developing a CBPR approach for health, describing a process for creating, maintaining a CBPR partnership: how to bring together partners, identify assets, select priority issues; challenges and benefits of conducting CBPR.
Social Determinants of Health Equity—Part 1	Course focused on describing health disparities/health inequities in the U.S.; the role of social and economic conditions in health disparities/inequities; describing an example of the application of CBPR to designing, implementing and evaluating intervention (e.g., program, policy, social change) research to address SDOH, and the implications of social and economic conditions for strategies to improve health/eliminate health inequities.
Social Determinants of Health Equity—Part 2	Course focused on describing health inequities in the U.S. and Mississippi, the role of social and economic conditions in health inequities; strategies that address social and economic conditions in order to improve health and eliminate health inequities.
Social Determinants of Health Equity Workshop for Community Health Advisors (CHAs)	Course focused on describing health inequities, discussing the role of social and economic conditions in health inequities, the implications of social and economic conditions for strategies to improve health and eliminate health inequities, and how CHAs can apply SDOH in their community work.
Advocating for Policy Change	Course focused on what is policy, difference between programs and policies; choosing strategies for winning policy change; power mapping, planning a campaign to change policy.
Taking Action for Change: SDOH and Policy Advocacy	Course focused on improving SDOH and equity, addressing health inequities in the Jackson Area, taking action for policy change, CBPR for policy change, using research for change to improve SDOH, action planning to address health inequities in the Jackson area and taking action and working together for policy change.

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
