# Peer review of "Twenty Years of Leading the Way among Cohort Studies in Community-Driven Outreach and Engagement: Jackson State University/Jackson Heart Study"

_ijerph, 2021, doi:10.3390/ijerph18020696_

Round 1

Reviewer 1 Report

The JHS study has been followed with great national interest, especially among the African-American community and fellow HBCU institutions.  Given the import of the study, it has also earned the attention of the public health and medical communities. This paper provides a concise overview of the journey and structure underlying a 20 year CVD community-based research project. The foundation and structure then allowed for garnering invaluable data and maintaining effective (sometimes challenging) relationships with the community and funders.  

The only question I asked is whether the two Social Determinants of Health Equity courses listed in Table 3 Jackson State University Jackson Heart Study (JHS) Community Outreach Center (CORC) 273 Trainings 2013‐2017 on page 8 were meant to have the same title even though the course description was slightly different for each. If the courses are intended to be listed in that manner (or perhaps reflect a two-part course), then the article is fine for publication.

Author Response

Reviewer 1

Comments and Suggestions for Authors

The JHS study has been followed with great national interest, especially among the African-American community and fellow HBCU institutions.  Given the import of the study, it has also earned the attention of the public health and medical communities. This paper provides a concise overview of the journey and structure underlying a 20 year CVD community-based research project. The foundation and structure then allowed for garnering invaluable data and maintaining effective (sometimes challenging) relationships with the community and funders.  

  1. The only question I asked is whether the two Social Determinants of Health Equity courses listed in Table 3 Jackson State University Jackson Heart Study (JHS) Community Outreach Center (CORC) 273 Trainings 2013‐2017on page 8 were meant to have the same title even though the course description was slightly different for each. If the courses are intended to be listed in that manner (or perhaps reflect a two-part course), then the article is fine for publication.

Authors’ Response:

Clarification was provided in Table 3 on page 8-- the two Social Determinants of Health Equity courses were modified to reflect Part 1 and Part 2.

Thank you very much for the comments/suggestions. We have tried to incorporate them to the best of our ability.

Reviewer 2 Report

Appropriateness of Citations:

Many of the citations do not seem to fit the contextual needs for cite. That is, the citations do not seem to directly support the text. Further, rather than citing seminal papers from the literature, many seem to simply be previous manuscripts published by the current authors. Certainly it is acceptable to cite one’s own manuscripts when the current study builds on your previous research, but not when they are not a good fit and not at the exclusion of other researchers work. Some examples include (but are not limited to, and so authors need to review all citations):  

Lines 49-50: Authors state: “Details of the JHS, its development, participants, and overall procedures are described in earlier publications [1]” however only one publication is cited. Either change to singular or cite additional manuscripts. Further, the citation is for and article that discusses the psychometric analysis of a specific tool, not overall procedures for the JHS. Surely there are more on topic manuscripts detailing the Jackson Heart Study’s clinical efforts to point readers to.

Lines 50-52: Authors state: “The public health burden caused by cardiovascular disease (CVD) continues to adversely affect individuals through cost, life expectancy, medical, pharmaceutical and hospital care, particularly in the case of African Americans [2]”. However, the authors only referenced citation is an 11 year old manuscript (again, authored by several of the current authors) focusing on the development of the Jackson Heart Study coordinating center (rather than one or more recent manuscripts focusing on the high disparities of the medical and financial impact of CVD on African Americans.

Lines 55-57: Authors state “Community engagement requires community partners to develop activities to enhance the quality of life for vulnerable populations, to reduce risk factors, and to decrease untimely sickness and death [4]”. Yet again, the single citation directs readers to another previous manuscript produced by the current authors.  

Lines 64-68: Again, only a single citation, this time specific to African American women only. Authors need to reference at least one more that is inclusive of males. Also, the sentence is long (consider splitting into multiple sentences) and somewhat confusing in that it is difficult to interpret. For example, the statement “…accelerated mistrust of research by African‐Americans” should specify “research conducted with African Americans” or “African American’s participating in research studies”.

Citations Needed:

Line 58-59: Statement needs one or more citations to substantiate.

Lines 59-61: Citations needed for statements about specific worries and also for statement about resulting health outcomes.

Materials and Methods:

Lines 71-73: The authors state that “This review enumerates the techniques that were utilized by the JSUJHSCORC to facilitate recruitment and retention of the JHS cohort…”. This directly implies to the reader that the CORC was actively involved in facilitating cohort recruitment and retention. However, there are some inconsistences that must be reconciled:

  • If the CORC was supporting recruitment to the JHS, the authors need to discuss what activities were conducted and the direct impact these activities had on cohort accrual rates and subsequent retention.
  • In Lines 76-77, the authors state that the CORC functioned during the period of 1999-2018, and that the “last contract period of the CORC (2013-2018)”. Implying there was a previous contract period. However, Table 2 states that the CORC did not form until 2013.

Lines 86-90: The authors discuss “Addressing the complexity of recruitment and retention…” and “Success for the JHS…”, leading to a discussion of the theoretical models the CORC implemented to accomplish or facilitate the JHS success’ in recruitment and retention. The Authors then cite the CCSM and the collaborative citizen science approach as the models used. However

  • The citation for CCSM is from 2018, which is after the end of the CORC’s participation in the JHS and which focuses on urban policy recommendation
  • The citation for the citizen science approach is from 2009 (well after study accrual was complete) and seems to focus on “Informal Science Education”.

Given the timing of the research models implemented by the CORC, the reader would be led to assume that models were applied retrospectively, rather than prospectively as the authors stated.

Lines 114-130: First, most of this paragraph (lines 116-125) are all one long and difficult to follow sentence (consider revising). Second, the authors state that “There were three iterations of the JHS efforts to refine the principles and levels of community engagement.” It would be useful to refer readers to Table 2 (timeline) here. Further, the CORC is indicated to be the 3rd iteration. Please explain to the reader how this aligns with the CORC supporting recruitment efforts.

Conclusions:

Line 497-498: The authors state that: “The work of the CORC is credited with facilitating recruitment of the largest cohort of African Americans to participate in a study of CVD.” This needs one or more citations to support the statement. Meaning, who credits the CORC with these accomplishments?

Line 499-502: Confusing statement, please revise. The authors indicate that the CORC is the “first and only center dedicated to community outreach and engagement…”. Is the JHS no-longer funded or does it no-longer have a community engagement center associated with it? A quick search would suggest that it does. The authors may want to discuss this newest iteration if it is already functioning. Perhaps as iteration 4?

Author Response

Reviewer 2

Comments and Suggestions for Authors

Appropriateness of Citations:

  1. Many of the citations do not seem to fit the contextual needs for cite. That is, the citations do not seem to directly support the text. Further, rather than citing seminal papers from the literature, many seem to simply be previous manuscripts published by the current authors. Certainly it is acceptable to cite one’s own manuscripts when the current study builds on your previous research, but not when they are not a good fit and not at the exclusion of other researchers work. Some examples include (but are not limited to, and so authors need to review all citations):  

Authors’ Response:

We appreciate the reviewer’s concerns and have added additional references that may help to support some of the content included. It is important to note that this work was intended to highlight approaches used by the JHS, represented by the co-authors, to conduct community outreach and engagement, and, as a result, the work of the authors over 20 years are presented to chronicle what was done and what was accomplished. We think the reviewer’s comments can add an extra dimension to our presentation. Thank you very much.

  1. Lines 49-50: Authors state: “Details of the JHS, its development, participants, and overall procedures are described in earlier publications [1]” however only one publication is cited. Either change to singular or cite additional manuscripts. Further, the citation is for an article that discusses the psychometric analysis of a specific tool, not overall procedures for the JHS. Surely there are more on topic manuscripts detailing the Jackson Heart Study’s clinical efforts to point readers to.

Authors’ Response:

We have inserted the following additional three citations relating to the details of the JHS, its development, participants, and overall procedures:

Sempos CT, Bild DE, Manolio TA. Overview of the Jackson Heart Study: a study of cardiovascular diseases in African American men and women. Am J Med Sci. 1999 Mar;317(3):142-6.

Crook ED, Taylor H. Traditional and non-traditional risk factors for cardiovascular and renal disease in African Americans: a project of the Jackson Heart Study investigators. Am J Med Sci. 2002 Sep;324(3):115.

Wyatt SB, Williams DR, Calvin R, Henderson FC, Walker ER, Winters K. Racism and cardiovascular disease in African Americans. Am J Med Sci. 2003 Jun;325(6):315-31. Review.

  1. Lines 50-52: Authors state: “The public health burden caused by cardiovascular disease (CVD) continues to adversely affect individuals through cost, life expectancy, medical, pharmaceutical and hospital care, particularly in the case of African Americans [2]”. However, the authors only referenced citation is an 11 year old manuscript (again, authored by several of the current authors) focusing on the development of the Jackson Heart Study coordinating center (rather than one or more recent manuscripts focusing on the high disparities of the medical and financial impact of CVD on African Americans.

Authors’ Response:

We have inserted the following three additional citations relating to recent manuscripts focusing on the high disparities of the medical and financial impact of CVD on African Americans.:

Mensah GA. Cardiovascular Diseases in African Americans: Fostering Community Partnerships to Stem

the Tide. Am J Kidney Dis. 2018 Nov; 72(5 Suppl 1): S37–S42. doi: 10.1053/j.ajkd.2018.06.026. PMCID: PMC6200348 NIHMSID: NIHMS1502869 PMID: 30343722

Wangui Odoi, E,,   Nagle, N,  Zaretzki, R,   Jordan, M,  DuClos, C,   Kintziger, KW. Sociodemographic Determinants of Acute Myocardial Infarction Hospitalization Risks in Florida. J Am Heart Assoc. 2020 Jun 2; 9(11): e012712. doi: 10.1161/JAHA.119.012712. PMCID: PMC7428988 PMID: 32427043

Carnethon, MR, Pu , J, Howard, G, Albert, MA, Anderson, CAM, Bertoni, AG, Mujahid, MS, Palaniappan, L, Taylor Jr, HA, Willis, M, Yancy, CW. Cardiovascular Health in African Americans: A Scientific Statement From the American Heart Association. Circulation. 2017;136(21), :e393–e423

  1. Lines 55-57: Authors state “Community engagement requires community partners to develop activities to enhance the quality of life for vulnerable populations, to reduce risk factors, and to decrease untimely sickness and death [4]”. Yet again, the single citation directs readers to another previous manuscript produced by the current authors.  

Authors’ Response:

We have inserted the following three additional citations relating to recent manuscripts focusing on Community engagement and community partners to develop activities to enhance the quality of life for vulnerable populations, to reduce risk factors, and to decrease untimely sickness and death:

Kruk, ME, Gage, AD, Arsenault, C, Jordan, K., Leslie, HH, Roder-DeWan, S. Adeyi, O, Barker, P, Daelmans, B, Doubova, SV, English, M, Elorrio, EG, Guanais, F, Gureje, O, Hirschhorn,LR, Jiang, L, d Kelley, E, Lemango, ET, Mohanan, J, Ndiaye, Y,  Norheim, K.  Reddy, S, Rowe, AK, Salomon, JA, Thapa, G,  Twum-Danso, NAY, Pate. M. High-quality health systems in the Sustainable Development Goals era: time for a revolution. The Lancet Global Health Commission. www.thelancet.com/lancetgh Vol 6 November 2018 e1196

Cyril S, Smith BJ, Possamai-Inesedy A, Renzaho AM. Exploring the role of community engagement in improving the health of disadvantaged populations: a systematic review.

Glob Health Action. 2015;8:29842. doi: 10.3402/gha.v8.29842. eCollection 2015.PMID: 26689460

Schroeder, K, Garcia, B,  Snyder Phillips, R,  Lipman. TH. Addressing Social Determinants of Health Through Community Engagement: An Undergraduate Nursing Course. J Nurs Educ . 2019 Jul 1;58(7):423-426.  doi: 10.3928/01484834-20190614-07. PMID: 31242312.  DOI: 10.3928/01484834-20190614-07

  1. Lines 64-68: Again, only a single citation, this time specific to African American women only. Authors need to reference at least one more that is inclusive of males. Also, the sentence is long (consider splitting into multiple sentences) and somewhat confusing in that it is difficult to interpret. For example, the statement “…accelerated mistrust of research by African‐Americans” should specify “research conducted with African Americans” or “African American’s participating in research studies”.

Authors’ Response:

 We have inserted the following additional citations that is inclusive of males, and the sentence has been modified according to the recommendation

Holt, CL,  Le, D, Calvanelli, J, Huang, J, Clark, EM, Roth, DL, Williams, B,  Schulz, E.

Participant Retention in a Longitudinal National Telephone Survey of African American Men and Women. Ethn Dis. 2015 Spring; 25(2): 187–192. PMCID: PMC4593062. NIHMSID: NIHMS725500

PMID: 26118147

  1. Citations Needed:

Line 58-59: Statement needs one or more citations to substantiate.

Authors’ Response:

We have inserted the following three citations to substantiate the statement:

 George S, Duran N, Norris K. A Systematic Review of Barriers and Facilitators to Minority Research Participation Among African Americans, Latinos, Asian Americans, and Pacific Islanders. American Journal of Public Health | February 2014, Vol 104, No. 2

Barrett NJ;  Ingraham KL ; Hawkins TV ;  Moorman PG. Engaging African Americans in Research: The Recruiter’s Perspective. Ethn Dis. 2017;27(4): 453-462. DOI: 10.18865/ed.27.4.453

Powell W, Richmond J, Mohottige D, Yen I, Joslyn A, Corbie-Smith G. Medical Mistrust, Racism, and Delays in Preventive Health Screening Among African-American Men. Behav Med. 2019 Apr-Jun;45(2):102-117. doi: 10.1080/08964289.2019.1585327. PubMed PMID: 31343960.

  1. Lines 59-61: Citations needed for statements about specific worries and also for statement about resulting health outcomes.

Authors’ Response:

We have inserted the following two citations to substantiate the statements about specific worries and also for statement about resulting health outcomes:

Noonan, AS , Velasco-Mondragon HE,  Wagner FA. Improving the health of African Americans in the USA: an overdue opportunity for social justice Public Health Reviews (2016) 37:12.  DOI 10.1186/s40985-016-0025-4

Taylor J. Racism, Inequality, and Health Care for African Americans. Report Health Care. December 19, 2019

Materials and Methods:

Lines 71-73: The authors state that “This review enumerates the techniques that were utilized by the JSUJHSCORC to facilitate recruitment and retention of the JHS cohort…”. This directly implies to the reader that the CORC was actively involved in facilitating cohort recruitment and retention. However, there are some inconsistences that must be reconciled:

  • If the CORC was supporting recruitment to the JHS, the authors need to discuss what activities were conducted and the direct impact these activities had on cohort accrual rates and subsequent retention.
  • In Lines 76-77, the authors state that the CORC functioned during the period of 1999-2018, and that the “last contract period of the CORC (2013-2018)”. Implying there was a previous contract period. However, Table 2 states that the CORC did not form until 2013.

Lines 86-90: The authors discuss “Addressing the complexity of recruitment and retention…” and “Success for the JHS…”, leading to a discussion of the theoretical models the CORC implemented to accomplish or facilitate the JHS success’ in recruitment and retention. The Authors then cite the CCSM and the collaborative citizen science approach as the models used. However

  • The citation for CCSM is from 2018, which is after the end of the CORC’s participation in the JHS and which focuses on urban policy recommendation
  • The citation for the citizen science approach is from 2009 (well after study accrual was complete) and seems to focus on “Informal Science Education”.

Given the timing of the research models implemented by the CORC, the reader would be led to assume that models were applied retrospectively, rather than prospectively as the authors stated.

Lines 114-130: First, most of this paragraph (lines 116-125) are all one long and difficult to follow sentence (consider revising). Second, the authors state that “There were three iterations of the JHS efforts to refine the principles and levels of community engagement.” It would be useful to refer readers to Table 2 (timeline) here. Further, the CORC is indicated to be the 3rd iteration. Please explain to the reader how this aligns with the CORC supporting recruitment efforts.

Authors’ Response:

As mentioned in the document, the unit that was instituted to conduct community engagement outreach, health education, and health promotion was the same unit organized and operated by Jackson State University. The main objective of this paper is to outline the activities that were conducted by the JSU community-focused unit and the direct impact these activities had on cohort accrual rates and subsequent retention. The name change throughout the years from 2000-2013 reflected modifications to the mission and objectives based on ongoing assessments/evaluations and recommendations from the OSMB. These modifications described in this account had the ultimate goal of perfecting the craft of recruitment and retention. The major goal of all of the activities of the JSU staff was to stimulate recruitment and retention in order to facilitate a successful research enterprise.

This account sought to provide the historical context for the successes of the JHS recruitment and retention strategies from 2000-2018 that was spearheaded by Jackson State University, and this historical account includes the JHS community outreach and engagement accomplishments that were highlighted in previous accounts. The impetus for the unit was to add to the science of community outreach and engagement and to provide current and prospective community engagement and outreach enthusiasts with a comprehensive account of the strategies used to build trust in an African American community with regards to participation in research and to align the interest of the recruited cohort with the commitment to ensure a successful research experience.

Lines 116-125 were modified according to the recommendations of the reviewer. Other edits have been made throughout the document to make the flow of the discussion easier to understand and in line with the suggestions.

Conclusions:

Line 497-498: The authors state that: “The work of the CORC is credited with facilitating recruitment of the largest cohort of African Americans to participate in a study of CVD.” This needs one or more citations to support the statement. Meaning, who credits the CORC with these accomplishments?

Authors’ Response

The following statement and citation was added in support of the statements made: The CORC efforts to maintain a viable cohort was successful in securing the highest retention rate among NIH-funded studies. Eighty-five percent of the cohort returned from Exam 1 to Exam 2. Reference #35 supported this statement: Crump, M.E.; Addison, C.C.;, Antoine-LaVigne, D; Walker, E.; Wilson, D.; Sarpong, D. Retention Patterns In The Jackson Heart Study: A Systematic Review. Paper presented at Seventh International Symposium on Recent Advances in Environmental Health Research. http://ehr.cset.jsums.edu/7cd/StudentPdf/Students%20Abstracts%2082.pdf. Retrieved December 23 2020.

Line 499-502: Confusing statement, please revise. The authors indicate that the CORC is the “first and only center dedicated to community outreach and engagement…”. Is the JHS no-longer funded or does it no-longer have a community engagement center associated with it? A quick search would suggest that it does. The authors may want to discuss this newest iteration if it is already functioning. Perhaps as iteration 4?

Authors’ Response:

The NHLBI contract to JSU for operating the CORC was not renewed in the contract period that began in August 2018 and extends to August 2023. There is no longer a functioning CORC, and this is an opportune time to detail, evaluate, assess,  and document  JSU’s history and accomplishments in the area of community engagement with NHLBI.

Authors’ Response:

Thank you very much for the comments/suggestions. We have tried to incorporate them to the best of our ability.

Reviewer 3 Report

The long-term success and impact of the Jackson State University (JSU) Jackson Heart  Study (JHS) Community Outreach Center (CORC) is a value to public health and community-based participatory partners so the challenges and successes of the journey can be understood and coalitions can create realistic expectations.  Many coalitions struggle to build relationships and sustain, and the JSU and JHS collaboration has made a substantial impact over time.

Many researchers might be starting to build capacity and unaware of the vision of long-term sustainability. The authors, provide a practice-based experience that impacts policy, research, institutional change, and training.

My recommendations are to reorganize the results in order of the six facet model of community engagement on line 437. The framework can be introduced earlier and be used to present the result in six categories. 

More discussion and examples are needed to describe the lessons learned from the challenges and conflicts in line 483-489. How authors navigated decision-making and voice is an essential process and CBPR principle.

Author Response

Reviewer 3

Comments and Suggestions for Authors

The long-term success and impact of the Jackson State University (JSU) Jackson Heart  Study (JHS) Community Outreach Center (CORC) is a value to public health and community-based participatory partners so the challenges and successes of the journey can be understood and coalitions can create realistic expectations.  Many coalitions struggle to build relationships and sustain, and the JSU and JHS collaboration has made a substantial impact over time.

Many researchers might be starting to build capacity and unaware of the vision of long-term sustainability. The authors, provide a practice-based experience that impacts policy, research, institutional change, and training.

My recommendations are to reorganize the results in order of the six facet model of community engagement on line 437. The framework can be introduced earlier and be used to present the result in six categories. 

Authors’ Response

Reorganization of the narrative has been made to reflect the suggestions. The framework was introduced earlier in accordance with this recommendation.

More discussion and examples are needed to describe the lessons learned from the challenges and conflicts in line 483-489. How authors navigated decision-making and voice is an essential process and CBPR principle.

Authors’ Response:

In addition to some realignment of text in the document, the statements below have been added to the Discussion section:

Barriers encountered have been overcome by targeted JSU JHS staff communication and debriefings, implementation of specific community activities to engage and include community members, awareness and sensitivity to the cultural and ethnocentric needs of the community, and the willingness to take time to explain study activities to participants and community members in a warm, respectful, and friendly atmosphere. The JSU JHS efforts on retention is a model for retention of African Americans in research. Similar retention activities can be applied by other researchers in minority settings and those investigators conducting research involving other racial/ethnic groups.

In the spirit of CBPR,  JSU JHS investigators combined personal qualities with relevant knowledge and experience to help participants and community members form opinions and make decisions regarding their continued participation in JHS health promotion and research activities. Through the  established  trusting, bonding relationship, the community partners understood that they were always afforded opportunities to interpret information to help them make the right choices for themselves and their families. To the JHS community members, their voice that echoed their emotions and attitudes became an important component of the decision-making process and an important tool of the JSU JHS team.

Authors’ Response:

Thank you very much for the comments/suggestions. We have tried to incorporate them to the best of our ability.

Reviewer 4 Report

Thank you very much for the article sent. I find the long experience of the project work very enriching, especially due to the emphasis on community participation.

Here are some comments that seek to improve the report of the document:

  • In general, reading is very "heavy", many repeated and disorganized ideas that make you lose the thread of reading. So also the wording needs to be more concrete and synthetic.
  • The structure of the article is very confusing. It may be better to adapt the structure to a review article (literary - no systematic review) that will help to better organize ideas
  • It seems to me necessary to organize the article describing the main strategies that have been implemented (considering all the phases of the project) and also show (if possible) results of each of these strategies, ideally with qualitative or quantitative results that support the aspects that are they mention.
  • The conclusions need to be more concrete and limited to the conclusions derived from the results

Author Response

Reviewer 4

Comments and Suggestions for Authors

Thank you very much for the article sent. I find the long experience of the project work very enriching, especially due to the emphasis on community participation.

Here are some comments that seek to improve the report of the document:

  • In general, reading is very "heavy", many repeated and disorganized ideas that make you lose the thread of reading. So also the wording needs to be more concrete and synthetic.
  • The structure of the article is very confusing. It may be better to adapt the structure to a review article (literary - no systematic review) that will help to better organize ideas
  • It seems to me necessary to organize the article describing the main strategies that have been implemented (considering all the phases of the project) and also show (if possible) results of each of these strategies, ideally with qualitative or quantitative results that support the aspects that are they mention.
  • The conclusions need to be more concrete and limited to the conclusions derived from the results

Authors’ Response:

Reorganization of the narrative was made to incorporate the suggestions and reduce repetition and confusion. Adjustments were also made to the conclusion to streamline the focus of the review.

In addition to some realignment of text in the document, the statement below has been added to the Discussion section:

Barriers encountered have been overcome by targeted JSU JHS staff communication and debriefings, implementation of specific community activities to engage and include community members, awareness and sensitivity to the cultural and ethnocentric needs of the community, and the willingness to take time to explain study activities to participants and community members in a warm, respectful, and friendly atmosphere. The JSU JHS efforts on retention is a model for retention of African Americans in research. Similar retention activities can be applied by other researchers in minority settings and those investigators conducting research involving other racial/ethnic groups.

Authors’ Response:

Thank you very much for the comments/suggestions. We have tried to incorporate them to the best of our ability.

Round 2

Reviewer 4 Report

Thank you very much for submitting a revised version of the manuscript and for incorporating some of the suggestions made previously. I believe that some of the suggestions made were not possible to implement possibly because it goes beyond the objective of this publication, although it would have been very enriching to incorporate results of each of these strategies, ideally with qualitative or quantitative results that support the aspects mentioned.

It is also appreciated that they have reviewed the writing of the manuscript and the bibliographic references.

I have no other additional comment to the manuscript.